# Potential Use of Compatible Osmolytes as Drought Tolerance Indicator in Local Watermelon (*Citrullus lanatus*) Landraces

**Lesego T. Sewelo [1], Kelebogile Madumane [1], Metseyabeng N. Nkane [1], Motlalepula Tait [2]**
**and Goitseone Malambane [1],***

[1]  Department of Crop and Soil Sciences, Botswana University of Agriculture and Natural Resources, Private Bag, Gaborone 0027, Botswana; 201700117@buan.ac.bw (L.T.S.); 201700403@buan.ac.bw (K.M.); 201200327@buan.ac.bw (M.N.N.)
[2]  Research Centre for Bioeconomy, Botswana University of Agriculture and Natural Resources, Private Bag, Gaborone 0027, Botswana; mtait@buan.ac.bw
*   Correspondence: gmalambane@buan.ac.bw

**Abstract:** Watermelons are one of the most important crop species, and they are enjoyed across the globe; however, the cultivation of watermelon commercial varieties in arid regions is challenging, as they are highly susceptible to water deficit. Conversely, their wild relatives and traditional landraces have shown a higher tolerance to water deficit, which makes them important study material. Therefore, this study was undertaken to evaluate the potential roles of two compatible osmolytes (citrulline and arginine) in the tolerance of local watermelon accessions to drought stress. Four commonly cultivated watermelon accessions were used in this study to evaluate their response when exposed to water deficit stress. The accessions were planted in stress boxes in the greenhouse and allowed to grow until the fourth leaf was fully open and then the water deficit stress was initiated by withholding water for a period of nine days, before rewatering for three days. Data and leaf samples were collected at three-day intervals. The common drought indicators that were assessed, like chlorophyll fluorescence, showed that Clm-08 (wild watermelon) had significantly different results when compared to the other accessions; the Fv/Fm values for days 3, 6, and 9 were significantly higher than those of the other accessions, while phiNPQ was higher in the Clm-08 with average values of 0.41 and 0.41 on days 6 and 9 of the drought stress, respectively. This suggests that the wild watermelon responded differently to drought stress when compared with the other accessions. Arginine and citrulline are important osmolytes that play an important role in stress tolerance, and the results of the current study correlate with the common physiological indicators. The expression pattern for both the biochemical and molecular analyses of the two compatible osmolytes was higher in Clm-08 in comparison with that of the other accessions. The gene expressions of the enzymes in the citrulline and arginine pathways were higher in Clm-08; *Cla022915* (CPS) recorded a 6-fold increase on day 6 and *Cla002611* (ASS) recorded an 11-fold increase. This suggests that citrulline and arginine play an important role in watermelon tolerance to drought stress.

**Keywords:** drought stress; climate change; citrulline; arginine; local watermelon accession

## 1. Introduction

Recently, crop production has shown a downward trend in most countries; this trend is aggravated by limited water resources and a reduction in arable land due to multiple factors, including climate change [1,2]. Climate change is the greatest challenge facing crop production, and it has had widespread negative impacts, such as drought stress, which reduces the productivity of crops [3,4]. Drought stress results in most crops failing to attain a maximum yield due to extreme temperatures and a shortage of soil moisture, which leads to a shortage of food [5]. Drought stress reduces crop productivity and yields by 30–50%, depending on the crop and the intensity of the drought stress [6]. These losses caused by

drought stress present a long-term challenge for food security as the human population continues to increase.

The greatest challenge facing the human species is population increase, which is causing an increased demand for food alongside the challenges that are brought about by climate change. As drought intensifies, there will be a shortage of food for the projected increase in the global human population. To achieve global food security by 2050, it has been predicted that crop production must increase in the range of 60–110%; if this is not achieved, extensive food insecurity will occur [7]. Therefore, there is a need to mitigate the effects of drought by evaluating the crops that have shown some level of resilience to this changing trend in the climate.

Drought comprises various components, such as high temperatures, high solar irradiance, low soil moisture, and low wind intensity. Of these components, low soil moisture is the most critical for crop growth due to its paramount importance in the physiological, biochemical, and molecular processes, which ultimately lead to yields [8]. Water deficit is responsible for extensive crop losses, which result in crops failing to reach their maximum yields [9]. These water deficit-related crop losses and poor yields have been attributed to negative changes in transpiration rate, translocation, stomatal conductance, and leaf-relative water content among various affected plant metabolisms [8,10]. Several strategies have been proposed as solutions to the challenge caused by drought stress; research into the breeding of tolerant crops has been put at the forefront as it promises to bring solutions [11]. For this to be successful, the screening and utilization of crops that are resilient to drought conditions (climate-smart crops) have key roles to play in the success of the research and breeding interventions.

Most of the wild relatives and local accessions of crops have been shown to withstand harsher environmental conditions than their cultivated relatives; this is mainly because of the ability of these species to adjust and survive under harsh conditions [12]. These important crop relatives of cultivated species constitute a critical gene pool for plant breeders, as they have traits that can be used to improve cultivated relatives [13]. Therefore, these crop relatives are ideal plant models for the study of abiotic stress because they possess important traits that can be used to provide longer-lasting solutions and to increase crop productivity during these challenging times [14]. One crop that has been shown to attract considerable consumer interest is the watermelon, which is commonly consumed as a snack or dessert in many households. However, watermelon is highly affected by drought stress [15,16]. In Botswana (and Southern Africa as a whole), the watermelon has a rich diversity that ranges from wild to cultivated varieties. The cultivation of these crop varieties is divided into two, with most of the commercial and improved subsistence farmers opting to cultivate improved cultivars, which have been shown to yield very well in good rainy years but have also been shown to be very susceptible to mild drought stress [17]. On the other hand, for several years, the traditional subsistence farmers have been cultivating local accessions that have been successful even in extreme drought years, thus making them important relatives of the watermelon family [18]. In this family, most importantly, is the wild watermelon, which is a common crop in the Kalahari Desert, where it grows and produces good sizeable fruit under harsh desert conditions [19,20], making it an important member of the family when it comes to research that is aimed at improving other relatives with regard to drought tolerance.

Proline is a common compatible osmolyte that has been widely studied and found to respond to drought stress in [21–24]; it accumulates during water stress and it plays an important role in scavenging ROS [25,26], maintaining homeostasis by reducing water potential in cells [27], and maintaining the driving gradient for water uptake [28,29]. Another important osmolyte associated with tolerance in plants is citrulline, which is a precursor of arginine [20]. Citrulline strongly accumulates in the leaves of plants in response to drought stress, and this suggests that citrulline plays a critical role in the active accumulation of solutes in the vacuoles (osmotic adjustment) and the scavenging of hydroxyl radicals, both of which contribute to the tolerance of some *Citrullus* species [19].

These species may act as a suitable plant model for the study of the underlying mechanisms, especially regarding the role of citrulline in association with crop tolerance to drought stress. Thus, the first objective of this study was to perform a comparative analysis of drought stress tolerance in the four most cultivated watermelon accessions in Botswana. The second objective was to elucidate the biochemical response of osmolytes, namely citrulline and arginine, and to document the transcriptional and biochemical correlation response patterns in cultivated and wild watermelon plants under water deficit conditions. We hypothesized that the wild and local accessions would possess a superior tolerance mechanism compared to that of the improved commercial cultivar. We also hypothesized that the accumulation of citrulline and arginine in *Citrullus* species is associated with tolerance to drought stress.

## 2. Materials and Methods

### 2.1. Experimental Site and Plant Materials

This study was conducted in a greenhouse (set at 30–35 °C and 60% humidity,) at Sebele BUAN (Botswana University of Agriculture and Natural Resources). The soils used in the study were collected from the BUAN gardens, where the soils are characterized as sandy loam and classified as Typic Haplustalfs [30]; no fertilizer was added during the experiments. The plant materials consisted of one wild accession, drought-tolerant watermelon (Clm-08), and three cultivated watermelon accessions, namely Clm-06, Clm-07, and Clm-09. Clm-08 wild watermelon plants have shown extreme drought stress tolerance and the ability to survive under harsh conditions in the Kgalagadi Desert [20]. The Clm-07 cooking watermelon plants have performed very well even in extreme drought years, which suggests that they have drought tolerance [31]. Clm-09, a local landrace watermelon, and Clm-06, a commercial (Tiger F1) hybrid, were also used in this study. The three watermelon accessions (Clm-07, Clm-08, and Clm-09) were obtained from the Botswana National Plant Genetic Resource Centre (BNPGRC) Seedbank, Ministry of Agriculture in Gaborone. The Clm-06 accession was purchased from commercial agricultural suppliers (Table 1).

**Table 1.** General information on the four *C. lanatus* (watermelon) accessions used in this study.

| Code | Accession Type | Source | Type of Use | Flesh Color | Reference |
|------|----------------|--------|-------------|-------------|-----------|
| Clm-06 | Hybrid | Agric shops | Fresh fruit | Red | [32] |
| Clm-07 | Landrace | BNPGRC | Cooking | Yellow | [33,34] |
| Clm-08 | Wild | BNPGRC | Feed | White | [35] |
| Clm-09 | Landrace | BNPGRC | Fresh fruit | Red | [36] |

### 2.2. Water Deficit Stress Experiment

The experiment was set up using a 2 × 4 factorial design, where the irrigation (well-watered and water-stressed) was factor A, while the watermelon accessions were factor B. All the seeds used in this experiment were established in a germination chamber to promote uniform germination. Soon after the radicle emerged, the plants were transplanted into stress boxes filled with sandy loam soil to a depth of 15 cm; the plants were sown at a depth of 3 cm and given the recommended agronomic management (irrigation, thinning, weeding) to establish the plant up to the 4th true leaf stage, which occurred at 3–4 weeks depending on the genotype. No fertilizer was added to the soil as the soil was loamy and from virgin land. The watermelon accessions were sown in two (2) drought stress boxes (box technique) following the methods in [37], where each watermelon treatment had six replicates of the main treatment, for a total of forty-eight experimental units. This study was repeated twice and the results averages after collection.

### 2.3. Irrigation Schedule and Drought Stress Initiation

The irrigation commenced immediately after planting, followed by a daily irrigation level of 10 L per box until the plants reached the 4th true leaf stage. Thereafter, the water deficit stress treatment was initiated by withholding irrigation for nine (9) consecutive days,

while the plants in the control box were continuously irrigated. Thereafter, rewatering was performed, during which the plants were watered with water at 10 L/day for the next 3 days before the final data were collected. To check the drought intensity, the soil moisture content was measured in 3 randomly selected plants for each accession at a depth of 10 cm using a calibrated MPM-160-B moisture probe meter (ICT International Pty Ltd., Armidalem, Australia). The soil moisture content was measured from the onset of stress imposition through to the termination of the experiment.

### 2.4. Physiological Data Collection

The fluorescence parameters were measured 4 h after sunrise using a MultispeQ V 2.0 m (PHOTOSYNQ INC., East Lansing, MI, USA). The measurements were conducted on the oldest leaf when the plants reached the 4th leaf stage, which happened an average of 4 weeks from emergence. These parameters included the chlorophyll content, the chlorophyll fluorescence ratio (Fv/Fm), the quantum yield of photosystem II (phi2), the ratio of incoming light that goes towards NPQ (PhiNPQ), and the ratio of incoming light that is lost via nonregulated processes (PhiNO). A portable SC-1 leaf porometer (METER Group, Inc., Pullman, WA, USA) was used to measure the stomatal conductance of three fully expanded leaves [38]. The relative water content (RWC) was measured by weighing fresh, turgid, and dried leaves using an analytic weighing balance. The turgid leaf weight was collected after the leaves were soaked in distilled water for 8 h in a dimly lit room at 25 °C and was recorded after 8 h. Subsequently, the turgid leaves were incubated at 80 °C for 48 h to determine the dry weight. Thereafter, the relative water content was estimated based on [39].

$$\text{RWC (\%)} = (FW - DW)/(TW - DW) \times 100$$

where FW is the fresh weight of the leaf, TW is the turgid weight, and DW is the dry weight.

### 2.5. Assays of Proline, Citrulline, and Arginine under Water Deficit Conditions

### 2.5.1. Sample Preparation

Leaf samples from all the accessions were collected at three (3)-day intervals (0, 3, 6, 9, and 12) from the onset of the water stress treatment. Leaves from random plants were detached 7 h after sunrise, snap frozen in liquid nitrogen, and stored at −80 °C until their use in molecular and biochemical analysis.

### 2.5.2. Extraction and Quantification of Citrulline and Arginine

Frozen leaf samples of approximately $0.2 \pm 0.01$ g were ground to a fine powder, extracted in 0.03 M of phosphoric acid (1.2 mL), and vortexed for 1 min. The samples were then sonicated for 30 min and incubated at room temperature for 10 min. The homogenized samples were then centrifuged for 20 min at 4 °C and $5700 \times g$. Approximately 1 mL of supernatant was filtered through Whatman filter paper and stored at −80 °C until use [40]. The quantification of the compatible osmolytes was performed using extracted crude extract, which was acquired using the ab273309 fluorometric citrulline assay kit and ab252892 fluorometric arginine assay kits (Abcam PLC, Tokyo, Japan) according to the manufacturer's instructions. The fluorescence was read using a Perkin Elmer LS 55 spectrofluorometer (PerkinElmer, Inc., Shelton, CT, USA) at an excitation wavelength of 535 nm and an emission wavelength of 587 nm in the endpoint mode. The capacity to estimate citrulline and arginine concentrations was calculated from the calibration curve made from citrulline and arginine standards and by plugging in the average fluorescence of the watermelon accessions as the x value [41].

### 2.5.3. Extraction and Quantification of Proline

This was determined as described in [42], where 1.25 g of ninhydrin was heated in 20 mL of 6 M phosphoric acid and 30 mL of glacial acid and stirred continuously until the mixture was homogenized. Approximately 0.5 g of fresh leaves from each sample was homogenized in 10 mL of 3% aqueous sulfosalicylic acid and centrifuged at 4000 RPM at a

temperature of 25 °C for 10 min, after which 2 mL of extract from each centrifuged sample was carefully placed in different test tubes. Approximately 2 mL of glacial acetic acid and 2 mL of ninhydrin acid were added to the test tubes and kept at 100 °C for an hour, after which the reaction was allowed to progress in an ice bath. In the reaction mixture, 4 mL of toluene was added, and the mixture was stirred for 15–20 s with a test tube stirrer. The mixture was carefully aspirated and then warmed to 25 °C. The absorbance was read at a 520 nm wavelength using a UV spectrophotometer. Sulfosalicylic acid was used as a blank, and a standard of a known concentration of proline was used to construct a standard curve. The absorbance values obtained were compared with the standard proline curve, and the results were expressed in $\mu$mol proline g$^{-1}$ FW (fresh weight).

### 2.6. Relative Expression of Citrulline- and Arginine-Related Genes under Drought Stress

This study used the watermelon water deficit-related reference genes *ClGAPDH* (*citrulas lanatus* glyceraldehyde-3-phosphatglyceraldehyde-3-phosphate dehydrogenase) and *ClActin* were selected from the work carried out in [43], and they were screened to confirm their stability across the genotypes and environment. The targeted genes of interest, namely *ORNITHINE CARBAMOYL-TRANSFERASE (Cla020781)*, *CARBAMOYL PHOSPHATE SYNTHASE (Cla022915)*, *ARGINOSUCCINATE LYASE (Cla022154)*, and *ARGINOSUCCINATE SYNTHASES (Cla002611)*, which are key genes associated with the citrulline and arginine pathways as shown in Figure 1, were used for the study. The known *Arabidopsis* and rice gene sequences were blasted against *C. lanatus* subsp. *vulgaris* cv. *97103* from the Cucurbit Genomics Database using TBLASTN [44]. The watermelon homologues were subsequently identified, and their sequences were used to design the primers using the Primer3Plus V3.2.0 online software tool (Table 2).

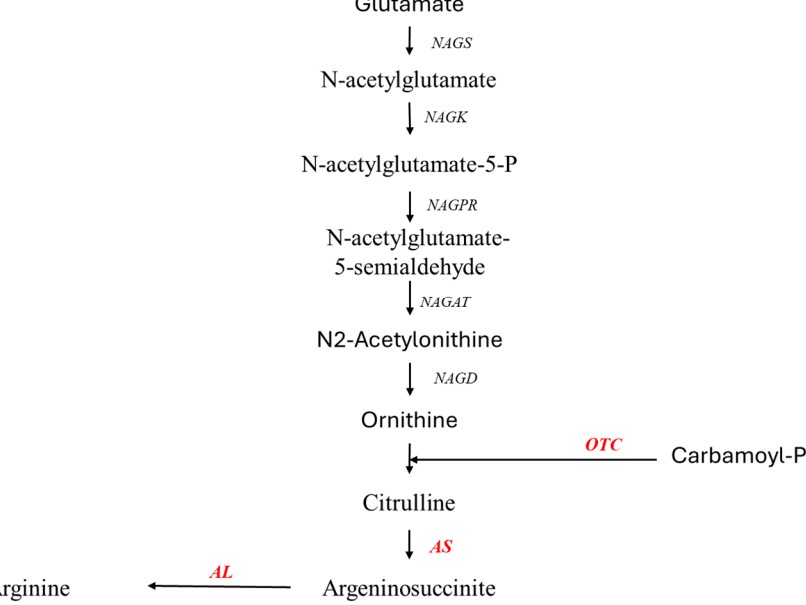

**Figure 1.** The citrulline and arginine biosynthesis pathways with the enzymes involved in the synthesis. Targeted enzymes in this study have been highlighted in red in the figure.

**Table 2.** Genes of interest (GOIs) encoding enzymes in the citrulline and arginine pathways and the primer sequences used to amplify the genes and reference genes in watermelon accessions.

| GOI | Enzyme | Pathway | Primer Sequences for GOIs |
|---|---|---|---|
| *Cla020781* | ornithine carbamoyl-transferase (OTC), | Citrulline | F_CTCTACTCACTTCTACTCCGGTACG<br>R_CTAAGATCTTCAAGAGGGTGGATTT |

**Table 2.** *Cont.*

| GOI | Enzyme | Pathway | Primer Sequences for GOIs |
|---|---|---|---|
| *Cla022154* | arginosuccinate lyase (ASL) | Arginine | F_ACATCTTCATGCACTAAACAGAGTG R_GATCCATTAGAGATGCTTGTGATCT |
| *Cla002611* | arginosuccinate synthases (ASS) | Arginine | F_GTGGAAGAAGCTCTACAAAGTCAAC R_AAGAGTGTCTTCTTCCTGGTTGTAA |
| *Cla022915* | carbamoyl phosphate synthase (CPS) | Citrulline | F_TCTCTACACTGTTCCTGAAAATTCC R_TGTTCTGACCAAAACCTAATCTCTC |
| Glyceraldehyde 3-phosphate dehydrogenase (GAPDH) | | | F_CTGGCAGTACTTTGCCAACA R_AGGATTGGAGAGGAGGTCGT |
| Actin | | | F_CCATGTATGTTGCCATCCAG R_GGATAGCATGGGGTAGAGCA |

Total RNA was isolated from the frozen samples using a Quick-RNA Plant Mini Prep Kit (Zymo Research Corporation, Irvine, CA, USA) according to the manufacturer's instructions. The RNA quality was checked via 1% agarose gel electrophoresis. A NanoDrop 2000 spectrophotometer (Thermo Fisher Scientific, Inc., Waltham, MA, USA) was used to measure the concentration and to confirm the purity of the isolated RNA at A260/280. cDNA synthesis was performed using a cDNA synthesis kit (New England BioLabs. Inc., Tokyo, Japan) with an oligo(dT) primer according to the manufacturer's instructions and amplified using ReadyMix 2 × PCR Master Mix Reagent (Thermo Fisher Scientific, Inc., USA) according to the manufacturer's instructions. The integrity of the synthesized cDNA was confirmed by amplifying the aliquots with the selected primers, running them through gel electrophoresis, and then viewing them via the gel documentation system to observe single amplified bands. The cDNA that did not produce bands was then both re-synthesized and amplified again for confirmation.

The synthesized cDNA was subsequently used as a template for gene expression analysis. RT-qPCR analysis was performed in a mixture containing 50 ng of cDNA template and 2 × Luna universal qPCR master mix (New England Biolabs, Inc., Tokyo, Japan) according to the manufacturer's instructions. The reactions were performed in a Bio-Rad CFX96 qPCR instrument (Bio-Rad Laboratories, Inc., Hercules, CA, USA), with two technical replicates for each of the three biological replicates. The negative controls were PCR mixtures without cDNA templates. A melting curve analysis was performed at the end of amplification to verify the specificity of the primers used [45]. The relative expression levels (Cq values) of each gene were normalized to those of the reference genes by taking an average of all the replicates. The relative expression levels were calculated using the $^{\Delta\Delta}$Cq (quantitative cycle) obtained from the Bio-Rad CFX, and the $2^{-\Delta\Delta CT}$ method was used to estimate the relative RNA expression [46].

*2.7. Data Analysis*

The recorded data were analyzed using the STATA computer analysis package, and the means were separated using the least significant difference (LSD) test at a probability level of 0.05.

**3. Results**

*3.1. Soil Moisture Content*

The soil moisture content percentage where control plants were grown ranged from 40.3% to 43.3% throughout the experiment. However, the difference in the moisture content of the well-watered plants between the sampling times was mostly insignificant. A sharp decrease in moisture content was recorded as early as the third day of moisture withdrawal, while a sharp increase was also observed 9 days after the rewatering of the stressed plants (Figure 2). Water deficit stress resulted in either an insignificant increase or no variation in the percentage moisture content throughout the experiment. The lowest moisture content

was recorded at 3.8%, which was recorded on day 9 of the stress treatment; thereafter, a sharp increase was observed post-rewatering.

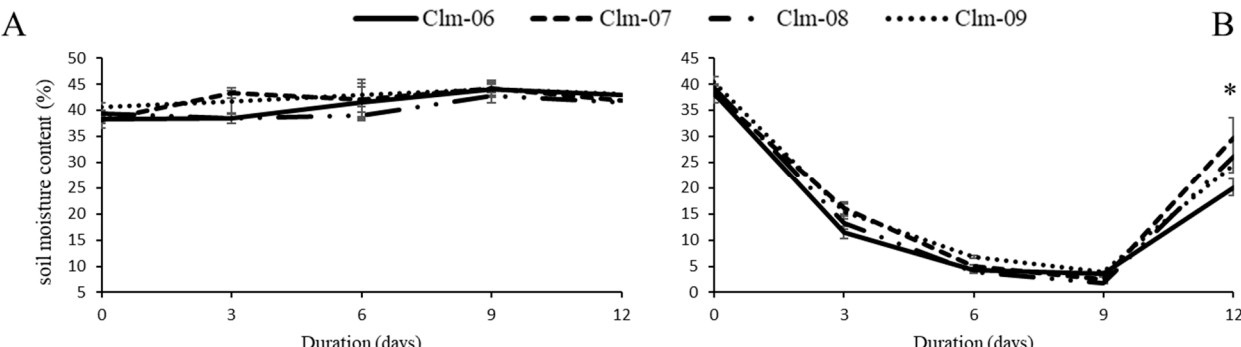

**Figure 2.** Soil moisture content measured at 3-day intervals with a soil moisture probe at a depth of 10 cm. Panel (**A**) represents the well-watered treatment, and (**B**) represents the water deficit stress treatment. The means are shown for *n* = 3, with error bars showing the standard deviations of the means. Statistical analysis was performed for each day of data collection; * significant difference at $p < 0.05$ per ANOVA performed using R v.4.3.1 software (2023).

*3.2. Physiological Parameters*

Upon the cessation of watering, plants were continually monitored for any signs of wilting, as wilting has been noted as one of the first visual signs of drought stress susceptibility. Plants that showed discoloration on the leaves were noted down as the first signs of wilting. The first signs of wilting were observed as early as 6 days post-water withdrawal, and this was recorded on Clm-06 (Figure 3). Clm-09 was not significantly different from Clm-06. The accession that did not show any signs of wilting at 9 days of stress was Clm-07, while Clm-08 did show milder signs of wilting at day signs.

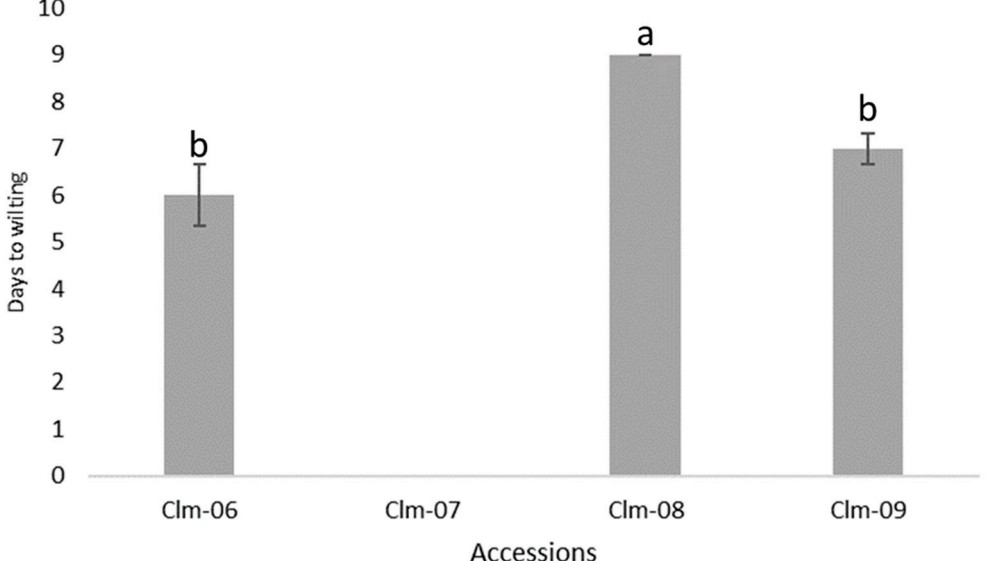

**Figure 3.** Effects of water deficit on leaf appearance on the studied accessions. The days were measured from the onset of the water deficit (day 0). The means are shown for *n* = 3, with error bars showing the standard deviations of the means. Multiple means were compared using Fisher's LSD test and were performed at a 95% significance level, and the significant differences among the means are highlighted by different letters on top of each bar plot.

The chlorophyll content in the well-watered plants demonstrated little to no variation for each accession for the duration of the study. However, Clm-08 showed a steady increase until the end of the study, where a significantly greater chlorophyll content was observed on days 9 and 12, recording averages of $49.6 \pm 2.2$ and $52.2 \pm 1.6$, respectively, under well-watered conditions (Figure 4A). The water deficit stress decreased chlorophyll content among the accessions. A marked decrease in chlorophyll content was observed in Clm-08, where a significantly lower chlorophyll content was recorded on days 6 and 9, with averages of $30.8 \pm 2.7$ and $23.2 \pm 1.9$, respectively; most interestingly, Clm-08 rapidly recovered after rewatering, during which a 50% increase was recorded, while the remaining accessions recorded a chlorophyll content increase of less than 50% after rewatering. The Clm-09 accession had a mild decrease in chlorophyll content, with averages of $52.4 \pm 3.7$, $45.5 \pm 1.7$, $44.0 \pm 2.8$, $46.1 \pm 2.6$, and $40.7 \pm 2.1$ on days 0, 3, 6, 9, and 12, respectively.

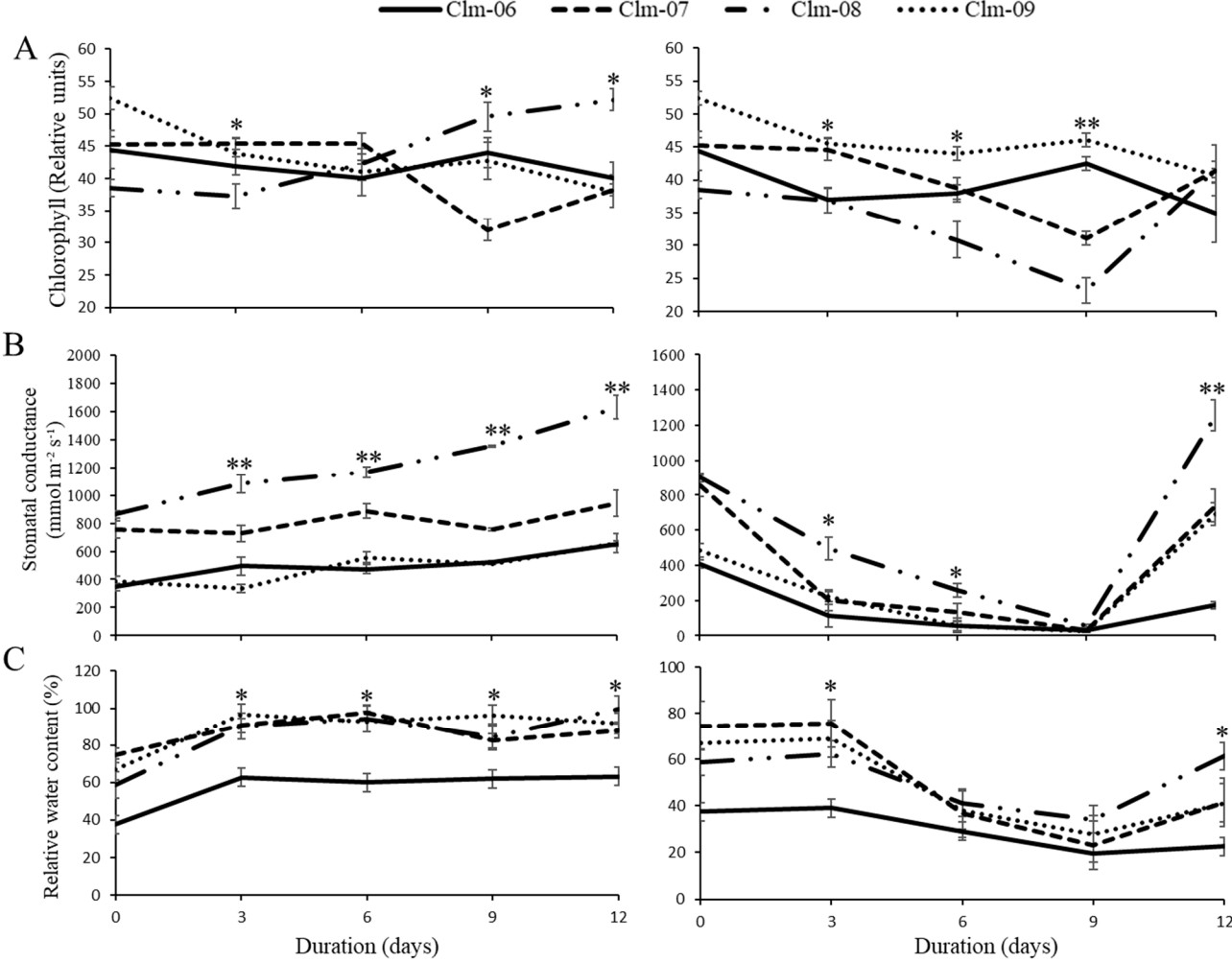

**Figure 4.** Effects of water deficit on the (**A**) chlorophyll content, (**B**) stomatal conductance, and (**C**) relative water content of the watermelon accessions as stress days progressed. The right panel represents water deficit-treated plants, while the left panel represents the well-watered plants. The means are shown for n = 3, with error bars showing the standard deviations of the means. Statistical analysis was performed for each day of data collection; * significant difference at $p < 0.05$, ** highly significant difference at $p < 0.05$ per ANOVA performed using R v.4.3.1 software (2023).

The highest stomatal conductance in the well-watered plants was recorded in Clm-08, followed by Clm-07, throughout the data collection period (Figure 4B), with Clm-08 reaching an average peak of $1630.3 \pm 271.6$ mmol m$^{-2}$ s$^{-1}$ on day 12. Moreover, there was no significant difference in the stomatal conductance between Clm-06 and Clm-09 in the well-watered plants. Water deficit stress resulting in a notable decrease in stomatal conductance was observed as early as 3 days and up to 9 days for all the accessions. Interestingly, Clm-08 had a significantly greater stomatal conductance on days 3 and 6 ($495.0 \pm 39.4$ and $259.0 \pm 6.3$ mmol m$^{-2}$ s$^{-1}$, respectively); however, on day 9, there was no significant difference in the other accessions. The stomatal conductance measurements after rewatering revealed a significant difference in the watermelon accessions. Clm-08 exhibited the highest average conductance at $1255.3 \pm 86.6$ mmol m$^{-2}$ s$^{-1}$, while Clm-06 exhibited a lower average conductance at $174.3 \pm 20.7$ mmol m$^{-2}$ s$^{-1}$.

The well-watered plants had similar RWC values for the entire data collection period, with only the Clm-06 plants having a significantly lower RWC for all the data collection days. Generally, water deficit stress resulted in a maintained RWC for the first three days for all the accessions. Consecutively, there was a decrease in RWC on day 6, and this response progressed throughout the entire sampling period. The lowest reduction in RWC was observed in Clm-06 on days 6 and 9, with the lowest RWC of 32% occurring on day 9 (Figure 4C). After rewatering, all the accessions showed a rapid increase, and the highest RWC was recorded in Clm-08 ($61.3 \pm 2.5\%$), while the lowest RWC was recorded in Clm-06 ($22.6 \pm 1.1\%$) on day 12.

### 3.3. Response of Fluorescence Parameters to Water Deficit

The quantum yields of PSII (Fv/Fm) and PhiNO decreased among the accessions in the early days of the water stress (Figure 5A,B). For Fv/Fm, the decrease in Clm-08 was milder than that in the other accessions, with an 18% decrease occurring from day 3 to day 9. Clm-08 had significantly greater Fv/Fm values of 0.7, 0.7, and 0.7 on days 3, 6, and 9, respectively. Three days post-rewatering, the Fv/Fm quickly recovered to near-normal values, as demonstrated on day 12. The second mildest (24%) decrease was observed in Clm-07, where significantly different values were also recorded on days 6 and 9. On day 3, the lowest Fv/Fm values were recorded for Clm-09 at $0.6 \pm 0.01$. On days 6 and 9, there was no significant difference between the lowest values recorded for the Clm-06 and Clm-09 watermelon accessions.

Exposure to drought stress resulted in an increase in PhiNPQ for Clm-07 and Clm-08, which both exhibited a twofold increase on day 9, while the two other accessions (Clm-06 and Clm-09) presented a less than onefold increase in PhiNPQ (Figure 5C). On day 3, there was no significant difference between Clm-07 and Clm-08 or between Clm-06 and Clm-09. However, significant differences were observed on days 6 and 9, when Clm-08 had higher values of $0.41 \pm 0.06$ and $0.4 \pm 0.1$, respectively. A decrease in all the accessions was observed post-rewatering, but there were no significant differences among the three genotypes, with only Clm-08 recording a significantly higher value of $0.1 \pm 0.02$ on day 12.

Under drought stress, the quantum yield of photosystem II (Phi2) decreased for all the accessions, and the decrease occurred as early as three days into the stress treatment (Figure 5D). The magnitude of decline varied among the accessions, with Clm-09 showing a milder decline, as it recorded higher values of Phi2 than the other accessions. The highest significant values of Phi2 were, thus, recorded in Clm-09 at $0.6 \pm 0.03$ and $0.61 \pm 0.01$ on days 3 and 6, respectively. Clm-08 had the lowest values but was significantly different only on day 6, when a Phi2 value of $0.4 \pm 0.01$ was recorded; however, the lowest value was recorded on day 9 even though it was not significantly different from the others.

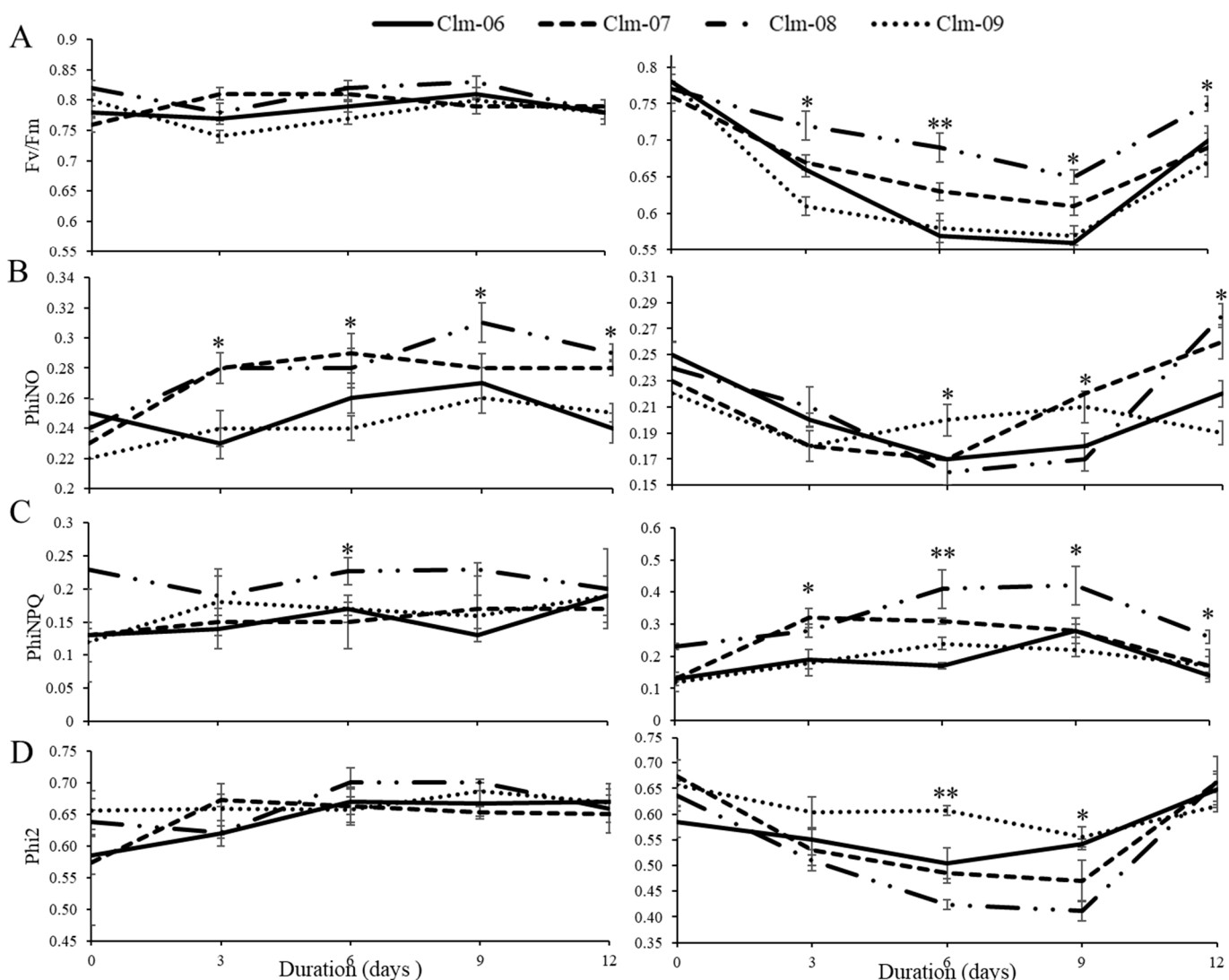

**Figure 5.** Effect of water deficit and excess light stress on watermelon accessions as determined by the (**A**) quantum yield of PSII (Fv/Fm); (**B**) the ratio of incoming light that is lost via nonregulated processes (PhiNO); (**C**) PhiNPQ, which is the ratio of incoming light that goes towards NPQ; and (**D**) the quantum yield of photosynthesis (Phi2). The left panel represents well-watered plants, whereas the right panel represents water-stressed plants. The means are shown for n = 3, with bars showing the standard deviations of the means. Statistical analysis was performed for each day of data collection; * significant difference at $p < 0.05$, ** highly significant difference at $p < 0.05$ per ANOVA performed using R v.4.3.1 software (2023).

*3.4. Quantification of Proline, Citrulline, and Arginine Concentrations*

Exposure to water deficit stress resulted in the accumulation of proline as the days progressed. All the accessions exhibited an increase even though the increase in intensity varied, with Clm-07 and Clm-08 exhibiting more intense increases of 83 and 68%, respectively, while the percent increase in Clm-06 and Clm-09 was less than 50% (Figure 6A). On day 3, there were no significant differences in the proline content recorded among the species. However, significant differences were first observed on day 6, when Clm-09 had a significantly lower proline content ($3.8 \pm 0.02$ µmol g$^{-1}$ FW). More significant differences were observed on day 9, when Clm-08 had a significantly greater proline content than Clm-07, at $7.9 \pm 0.01$ µmol g$^{-1}$ FW and $6.9 \pm 0.23$ µmol g$^{-1}$ FW, respectively. After rewatering, all the accessions exhibited a significant decrease in proline content, which was lowest in Clm-09 ($3.5 \pm 0.24$ µmol g$^{-1}$ FW) and highest in Clm-08 ($6.2 \pm 0.033$ µmol g$^{-1}$ FW).

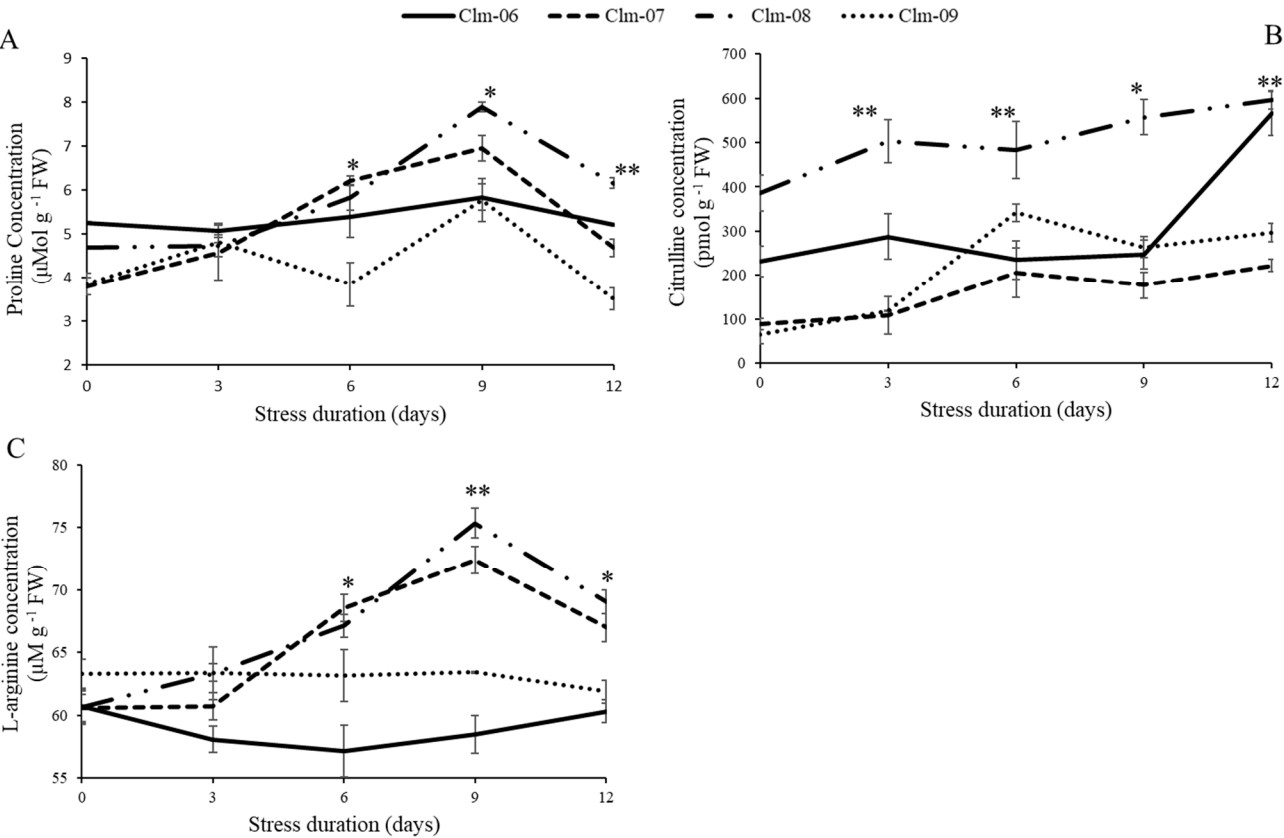

**Figure 6.** Three osmolytes related to the drought response evaluated in this study: (**A**) proline, (**B**) citrulline, and (**C**) arginine concentrations in water-stressed watermelon accessions (0–9 days) and during the post-stress rewatering period (day 12). The means are shown for n = 3, with bars showing the standard deviations of the means. Statistical analysis was performed for each day of data collection; * significant difference at $p < 0.05$, ** highly significant difference at $p < 0.05$ per ANOVA performed using R v.4.3.1 software (2023).

The citrulline content in the water-stressed plants resulted in an increase in all the accessions throughout the stress days (Figure 6B). Clm-08 had a significantly greater citrulline content on three consecutive stress days (days 3, 6, and 9), with the highest value recorded being $556.9 \pm 39.6$ pmol g$^{-1}$ FW. For the other three accessions, significant differences were observed on day 3, when Clm-06 recorded a greater citrulline content ($287.69 \pm 52.31$ pmol g$^{-1}$ FW), and on day 6, when Clm-09 recorded a greater citrulline content ($231.5 \pm 35.3$ pmol g$^{-1}$ FW). All the accessions continued to accumulate citrulline even after rewatering, as shown by the upward trend in the values recorded on day 12, that is, 3 days after rewatering.

The arginine concentration in Clm-09 was almost constant throughout the stress treatment, as shown in Figure 6C. Specifically, the concentration in Clm-06 decreased slightly on day 3 and recovered to almost normal levels on day 6; then, it remained almost constant. The arginine levels were greater in Clm-08 and Clm-07 than in the other strains on most of the stress days. The concentrations of Clm-08 and Clm-07 were not significantly different on any of the stress days except on day 9, when Clm-07 had a significantly greater level of arginine ($75.3 \pm 1.2$ μM g$^{-1}$ FW). After rewatering on day 12, the arginine concentration in all the crops slightly decreased, but there were no significant differences between Clm-08 and Clm-07.

### 3.5. Expression Analysis of Target Genes Involved in the Citrulline and Arginine Pathway

The expression levels of the genes in the arginine pathway (Figure 1) were quantified with specific primers, as shown in Table 2. The results revealed a greater relative expression

of Cla022915 in Clm-07 and Clm-08 on days 3, 6, and 9 than in the other two accessions (Figure 7A). On day 6, Clm-08 had a peak expression of the mRNA, showing a sixfold increase that was higher than the expression for other water stress days in the same accession. Clm-07 had its peak on day 9, recording a sevenfold expression. Clm-09 and Clm-06 exhibited less than a onefold increase in expression throughout the stress treatment days.

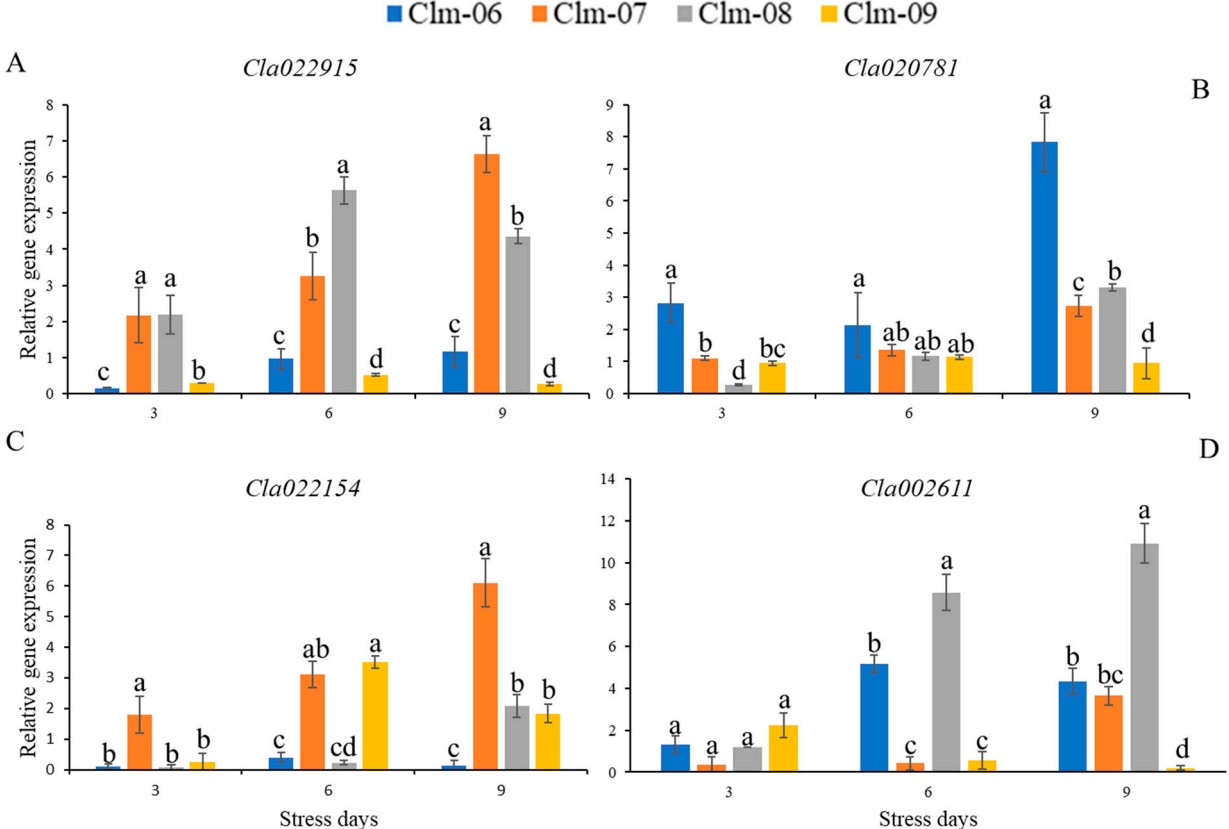

**Figure 7.** Relative expression of 4 genes associated with arginine, Cla022915 (**A**) and Cla020781 (**B**), and with citrulline accumulation, Cla022154 (**C**) and Cla002611 (**D**), relative to the average expression of two reference genes (actin and GAPDH) in watermelon leaves subjected to water deficit stress. The relative expression levels are presented as the means ± standard deviations and were calculated from three biological and two technical replicates following the $2^{-\Delta\Delta CT}$ method. Multiple means were compared using Fisher's LSD test and were performed at a 95% significance level, and the significant differences among the means are highlighted by different letters on top of each bar plot.

The highest expression of Cla020781 was recorded on Clm-06 where an 8-fold increase was noted on day 9 of the water deficit stress as compared to the other accessions that recorded a lower fold increase (Figure 7B). The second and third highest expressions on day 9 were recorded on Clm-08 and Clm-07 with 3.5- and 3-fold increases, respectively. Even though the expression levels of Clm-08 and Clm-07 were not the highest, they exhibited a steady increase from day 3 to day 9 of the stress treatment. The Clm-09 expression did not change on days 3 or 6, but a slight decrease was observed on day 9.

The Cla022154 expression increased steadily in the Clm-07 plants as the stress progressed; a twofold increase was recorded on day 3, and a sixfold increase was observed on day 9 as compared to Clm-09 plants that exhibited increased expression on day 6, after which the expression decreased significantly (Figure 7C). No increase in expression was observed in Clm-08 on day 3 or 6, but a slight increase was observed on day 9 (twofold increase), whereas the expression in Clm-06 was very low compared to that in the other accessions.

The Cla002611 expression steadily increased in Clm-08 after 3, 6, and 9 days of stress, with 1.5-, 8.5-, and 10.9-fold increases, respectively, whereas in other accessions like Clm-07, the expression increased by less than 1-fold initially and a 3.7-fold increase was recorded on day 9 (Figure 7D). The expression in Clm-09 showed a continuous decrease as water deficit stress days progressed as compared to the expression in other accessions, which showed a steady increase even though the magnitude varied.

## 4. Discussion

The moisture content of the soil upon water-stressed treatments declined to about 13% on day 6, and this shows that the plants were under water deficit stress, which triggered most of the response mechanisms like the accumulation of citrulline and proline, the quenching of excess light energy, and the reduction in stomatal conductance. It has been shown that plants begin wilting at a moisture content of around 10% [47]. For the watermelon accessions studied, early signs of wilting were observed in Clm-06 and Clm-09, while mild wilting was observed in Clm-8, and no wilting was observed in Clm-07. This method could be suggested for the preliminary screening of genotypes as their response can be determined easily by the appearance of the plant [48]. Plants are known to modify their morphology and their physiological and molecular functions in response to reduced soil moisture but the variation is influenced by the growth stage and by the species. These changes affect most of the biochemical processes of the plants, and in most cases, the effect is negative, such as a reduction in the photosynthetic capacity of the plant [49].

Drought stress is a critical environmental stress that limits the productivity of most crops and affects many morphological, physiological, biochemical, and molecular processes within plants. The effect of drought not only varies between species but also within species, and different genotypes have shown different response patterns [50,51]. Thus, it is important to evaluate this wild species to determine the mechanisms that aid in drought tolerance. One of the early physical signs of drought stress is a change in leaf color from green to yellow, indicating a loss of chlorophyll pigment, and the results of our study showed that the watermelon plants under drought conditions experienced a decrease in chlorophyll content. The decrease was more significant in the Clm-06 and Clm-09 accessions than in the other accessions, which suggests that these two accessions are susceptible to drought, as chlorophyll has been used as a tolerance indicator, and plants that are able to maintain their chlorophyll content are regarded as tolerant [52].

Stomatal conductance regulation is an important mechanism that plays a significant role in plant responses to water deficit; it measures the stomatal opening degree and is an estimate of the transpiration rate and gas exchange through leaf stomata [53]. In the present study, Clm-08 exhibited significantly greater stomatal conductance during water deficit stress than the other plants, except on day 9, when no significant difference was observed. Even though this accession is known to be drought tolerant, its closure was delayed, suggesting that other tolerance mechanisms like minimizing the biochemical reactions help the plant retain moisture slightly longer even when the stomatal conductance is high during water deficit stress [30]. Similar results were observed by the authors of [54], who reported decreased stomatal closure in drought-tolerant plants and suggested that antioxidant activities played a role in plant tolerance. The stomatal conductance of all the watermelons increased rapidly after rewatering, and these findings are consistent with the findings of [55], in which it is reported that water re-supply after stress increases stomatal conductance.

The RWC, which is also an important indicator of the water status of plants, has been widely used as an indicator of drought stress tolerance. Plants that can maintain their internal moisture longer under drought conditions have a chance of survival as their internal moisture helps to maintain plant status [56]. Interestingly, even though Clm-08 had higher stomatal conductance, it had a significantly greater RWC at day 9, suggesting that the accession managed to accumulate compatible solutes that aided in maintaining RWC, thus aiding drought tolerance [34,54]. Most importantly, the reversibility of RWC

after rewatering was observed in all the accessions, though at different magnitudes. This was corroborated by the authors of [57], who noted that this dehydration is often reversible when the plant has not reached its wilting point.

Significant differences in chlorophyll fluorescence parameters were observed between the well-watered and water-deficit-stressed plants and between the accessions subjected to stress treatment (Figure 4). Chlorophyll fluorescence parameters have been reliably used to evaluate the integrity of the photosynthetic apparatus of a plant during drought stress in various species [58,59]. In the present study, Clm-09 plants had a greater chlorophyll content on day 9 of water deficit stress, suggesting that the accession presented reduced pigment photo-oxidation and chlorophyll degradation [60]. After rewatering, there was an increase in chlorophyll content in all the accessions except for Clm-09, where a decrease was observed; this difference may be attributed to the remobilization of nitrogen from senescing leaves to newly formed leaves after rewatering [61].

An increase in Phi2 was observed in all the well-watered plants, whereas a decrease in Phi2 was observed in all the accessions under water deficit stress. The lowest values were recorded on day 9 of the stress, suggesting a reduced efficiency of the excitation energy capture in Phi2; this is a response mechanism to modify the response of Phi2 to downregulate the photosynthetic electron transport to match a decreased carbon dioxide assimilation [62]. This is supported by the authors of [63], who stated that Phi2 decreases with the increasing intensity of water deficit stress, as observed in this study. Furthermore, drought stress decreased Phi2 in the maize hybrids 30Y87 and 31888, suggesting that the photoinhibition of PSII occurred [8]. A slight increase in Phi2 values was observed in all the watermelons after rewatering, which suggests that water is critical for the generation of necessary electron and proton carriers in the electron transport chain for the overall process of photosynthesis [64].

A decrease in Fv/Fm was observed in all the accessions under water deficit stress treatment, with Clm-09 and Clm-06 exhibiting the lowest values, suggesting extreme desiccation in these accessions; it has been reported that plants respond to extreme desiccation [65] and that the responses are milder in drought-tolerant genotypes [66,67], as shown in the two accessions (Clm-07 and Clm-08) in this study that are most likely to be tolerant. The lowest values for all the accessions were recorded on day 9, which was in line with the findings by the authors of [68], who reported that Fv/Fm decreases significantly with increasing exposure to drought stress.

The results showed that PhiNPQ increased as the drought progressed, with Clm-08 recording the highest on day 9, suggesting that this watermelon could dissipate excess light energy, thus aiding in the photoprotection of PSII [19,69]. These results agree with the findings that state that the contribution of NPQ to the photoprotection of Phi2 differs according to the species or cultivar of the same species [70]. Additionally, Bashir et al. [8] reported results showing an increase in PhiNPQ that were like those in the present study in the drought-sensitive hybrid P3939 maize. A decrease in PhiNPQ in stressed plants occurred after rewatering, indicating improved Phi2 activity and overall photosynthetic capacity [71].

Osmolytes are compatible solutes that accumulate rapidly in plants in response to abiotic stresses to aid in the maintenance of osmotic balance [72]. When plants are exposed to water deficit, they accumulate osmolytes such as proline, glycine betaine, citrulline, and arginine to confer oxidative stress [73]. Proline is the most common osmolyte in water-stressed plants, and its accumulation is attributed to its biosynthesis activation and the inactivation of its degradation [74]. In the present study, the proline content of the water-stressed plants increased gradually at different magnitudes throughout the duration of the stress treatment, with Clm-07 and Clm-08 exhibiting the greatest accumulation. A greater accumulation of proline suggested the inactivation of proline degradation under water stress [75]. After rewatering, the proline content decreased in all the watermelon accessions, suggesting that proline plays a role in osmotic adjustment and structural protection [76]. In support of the findings of our study, [77] reported that the application of exogenous proline

maintained the turgidity of stressed barley and wheat plants, indicating that proline plays a role in osmotic balance.

Citrulline synthesized in the chloroplast is an intermediate nonessential amino acid in the arginine pathway that is involved in the maintenance of cellular osmolarity during abiotic stress [78]. In the present study, citrulline accumulation increased with the increase in days of stress, and this increase was continuous even after rewatering. The citrulline concentration in Clm-08 was greater after three consecutive stress days, indicating that citrulline accumulated dramatically in the leaves of this accession compared to the leaves of the other watermelon accessions, and it has been stated that the citrulline content varies with cultivar, genotype, and ploidy [20,79]. The increase in the citrulline concentration as drought stress progresses could suggest that citrulline plays an active role in the tolerance of wild watermelon [20]. Additionally, the accumulation of citrulline in watermelon leaves reportedly increases the antioxidative potential of leaves to protect cells from oxidative stresses [20]. This was also confirmed via the upregulation of CPS (Cla022915) and OTC (*Cla020781*), which are enzymes along the citrulline synthesis pathway that are activated by drought stress and are important in regulating the synthesis of citrulline [78,79] and shows that the wild watermelon accumulates citrulline in its leaves and stems as a tolerance mechanism [80]. The variations in the citrulline accumulation in the accessions studied can be partly attributed to the response to water stress as it has been suggested that water activity plays a role in the differences within species [40].

The concentration of arginine increased as the drought progressed but declined after rewatering in all the accessions. The highest concentrations were observed in Clm-08 and Clm-07, suggesting that these watermelons have a mechanism related to the arginine accumulation that aids them in coping with water deficits [74]. The increase in arginine content suggested that its synthesis increased to prevent osmotic stress [81–83]. This was confirmed by [84–86] after the application of exogenous arginine enhanced survival in tomato and maize plants under drought stress. The upregulation of ASS (*Cla002611*) and ASL (*Cla022154*), which are enzymes along the arginine synthesis pathway, further suggests that they have a role in drought stress tolerance. The accumulation was significantly expressed, suggesting that the tolerant accessions are Clm-07 and Clm-08. This was further corroborated by the authors of [87], who noted higher arginine expression in tolerant species under drought conditions.

## 5. Conclusions

The study showed the differences between the commonly cultivated watermelon accessions in Botswana and Southern Africa. The results showed the importance of comparing local and domesticated watermelon accessions as there was variation observed in the responses of the studied accession. Clm-07 and Clm-08, which are the cooking and wild watermelons, showed a greater tolerance to drought stress than the cultivated species; these two species are mostly neglected in the wild or treated as weeds on traditional farms. Thus, more attention should be focused on their tolerance mechanisms, which can be harnessed and used to improve the susceptible accessions. Importantly, this study confirmed the role of the two compatible osmolytes in the tolerance mechanisms of the watermelon. The accumulation of compatible osmolytes that have been suggested to play a role as an antioxidant and adding plant maintain internal moisture was visible in Clm-07 and Clm-08, which had a higher accumulation of this osmolyte and were noted to be drought tolerant. Thus, we can confirm that citrulline and arginine play an important role in drought tolerance and can be used as critical indicators that can be used to select the tolerant accession within the species. The continued screening and identification of the molecular and biochemical processes that play a significant role in climate-smart crops have become significant issues in the pursuit of food security. When identified, these mechanisms can be harnessed and used to improve important drought-susceptible species through the introgression of these important traits and the use of biotechnology tools.

**Author Contributions:** Conceptualization, L.T.S. and G.M.; methodology, L.T.S. and G.M.; software, L.T.S.; validation, L.T.S. and G.M.; investigation, L.T.S., K.M. and M.N.N.; resources, G.M.; data curation, L.T.S., K.M. and M.N.N.; writing—original draft preparation, L.T.S.; writing—review and editing, G.M. and M.T.; supervision, G.M. and M.T.; project administration, G.M.; funding acquisition, G.M. All authors have read and agreed to the published version of the manuscript.

**Funding:** This research was funded by the SG-NAPI award supported by the German Ministry of Education and Research, BMBF through UNESCO-TWAS (4500454040), and the Botswana University of Agriculture and Natural Resources.

**Data Availability Statement:** Data are contained within the article.

**Conflicts of Interest:** The authors declare no conflicts of interest.

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
