# Peer review of "Potential Use of Compatible Osmolytes as Drought Tolerance Indicator in Local Watermelon (Citrullus lanatus) Landraces"

_horticulturae, doi:10.3390/horticulturae10050475_

Round 1
Reviewer 1 Report
Comments and Suggestions for Authors
The manuscript tried to explore the responses of watermelon to drought stress. But, I think the results are incomplete and uninteresting, as the functions of the compatable osmolytes in drought stress are well known, have no need to further explore. Also, the results have some problems, such as:
(1) soil mosture content in Fig 1 first decreased and then increased at 12 d, it's not understandable.
(2) The change trend of the measured indicators during drought stress in the four watermelon accessions are different, sometime higher in one accessionm, but sometimes in other accession. not understandable.
(3) the author want to try to investigate the molecular mechanisms by examing four gene expressions. But the gene expression in four accessions are also different, can't make a conclusion. Also, the chosen four genes are not representative for osmolytes.
So, in my opinion, the manuscript should concentrate on two constract watermelon accessions with obviously different drought resistance, and try to explore their differences in response to drought by physiology and transcriptome analysis, not be confined to osmolytes, and no need to studied four accessions with no representativeness.
Author Response
Dear reviewer
We would like to extend our gratitude for your time and valuable comments that are aimed at helping us improve our manuscript.
We have addressed the comments and suggestions and also responded to the comments (attached)
Kind Regards

Reviewer 2 Report
Comments and Suggestions for Authors
The work is in an appropriate area and it is valuable to see land races v commercial lines being compared. Generally well written- the abstract given in the email differs from that in the paper ( paper version is much better)
Several sticky notes show places for comment or concern. In many cases this is to add more information into text rather than just being general.
These sticky notes itemize what could be changed in a revision.
A general problem is lack of labels on y axes or showing which is a b c d
this means one could not go back and forth between text and figures. Also would suggest that the line graphs should be bar charts there is no evidence how the lines join up between points
Adding information on proline into the abstract would be valuable- it is in results and discussion sections Also there are the CHO agents trehalose and sucrose- the decision on which osmolyte can rest with the tissue under examination and growth conditions
i like the suggestion that the assay of the osmolytes could be involved in screening for drought resistant lines (at gene or metabolite level)

Comments on the Quality of English Languagegenerally OK
flows well organization fine
Author Response

(The authors gave the same response as above.)

Reviewer 3 Report
Comments and Suggestions for Authors
The article follows the scientific method, however the novelty is low, as the role of citrulline has been already demonstrated on wild and commercial watermelon accessions. Authors must stress the novelty of their study. I think that the novelty may be related to the use of two additional “landraces”; however, a better description of these materials must be provided: are they registered/deposited somewhere? Also, authors must clearly describe what is already known and what they found for the first time. It is imperative to stress the novelty of this study. If possible, it would be also important to add some growth and production data.
The whole English language requires extensively revision by a native speaker to improve the readability, as it is wordy in some parts, and some sentences are constructed strangely. Examples were highlighted on the text, and explained in the abstract. Also, the statistical analysis must be improved, using a post-hoc test to effectively compare the accessions. This will also change the description and interpretation of the results.
The discussion is superficial and repetitive, as the same information is repeated several times, but no deeper explanation on mechanisms are provided. Authors often cite “mechanism” without going deep on which mechanism and how it works. Thus, the discussion must be improved.
Further comments were made on the text, and are listed below:
1. Title: As the osmolytes are already known to function on drought tolerance, I suggest to change the focus from osmolytes to the landraces used, if they were the novelty in your study.
2. The abstract lacks important information, such as the experimental description (what were the treatments, varieties, drought stress conditions, field/greenhouse conditions, plant age?)
3. Several sentences in the introduction are lacking references (see lines 35, 39, 58, etc).
4. How the drought stress affects plant metabolism? Please, better describe on introduction. Also, how does the osmolytes function in increasing drought tolerance? Please describe in the introduction.
5. Why did the authors specify only two osmolytes in the abstract, if proline was also evaluated? Also, why authors did not analyze the expression of genes related to proline?
6. The decimal separator must be changed from comma (,) to point (.) on figure 3.
7. Some results are misunderstood, as authors say that no differences among accessions were found for stomatal conductance under drought (line 282), but the Figure 2 clearly shows differences. Another example can be found in line 325.
8. Subjective terms that are used must be avoided, such as “least significant” (line 315), and “relatively similar” (line 21). Authors should go directly to: decreased or increased. It occurs in the entire results section. Please correct.
9. Authors should be consistent on how they call the treatments and variables throughout the manuscript. I suggest using well-watered instead of “moisture deficit” or “control”.
10. Authors must apply a post-hoc test, such as Tukey’s test, to identify which of the accessions were higher or lower than others. In the way the statistical analysis is presented, it only shows accessions were different, but it does not allow to known each was higher or lower. This will also change all the results writing. Please correct accordingly.
11. Please, express proline, citrulline and arginine contents in units of fresh or dry weight (for example, µmol g-1 fresh weight), not per volume. Correct these units in the figure 4.
12. Some figures are called wrongly (Figure 4 is called as 5) or are not called in the text (Figure 5). Please, revise the calling of all figures on the text.
13. In the discussion, authors mention a result that was not addressed on the results section (wilting) (line 425). Please, add this to results section or remove from discussion.
14. Why the plant growth and production were not evaluated?? These results are extremely important. They would be important to determine which accession is better for farmers, as the higher tolerance solely do not represent an advantage for farmers. They need productive responses too.
15. A correlation analysis or multivariate analysis should be added, as it would be important to check the relation among the tolerance responses (higher photosynthetic capacity, chlorophyll content, and relative water content) with the osmolytes contents and gene expression. Also, these analyses would help to discriminate accessions.
16. The role of citrulline on drought stress tolerance of wild watermelon accessions from Botswana were already assessed by Kawasaki et al. 2000 (doi: 10.1093/pcp/pcd005) and Akashi et al. 2001 (doi: 10.1016/S0014-5793(01)03123-4). Thus, what is the novelty of your study?
17. Conclusions should respond the hypothesis instead of being a summary of results. Please, revise it.

Comments on the Quality of English LanguageThe whole English language requires extensive revision by a native speaker to improve the readability, as it is wordy in some parts, and some sentences are constructed strangely. Examples were highlighted on the text, and explained in the abstract.
Author Response

(The authors gave the same response as above.)

Round 2
Reviewer 1 Report
Comments and Suggestions for Authors
I see that the author is trying to improve the quality of the paper, even if the paper is still lacking in novelty. If the editor finds the content suitable for publication, I agree to publish it as it is.
Author Response
We would like to take this opportunity to thank the editor and the reviewers for taking their valuable time to review our manuscript and make comments that will help in improving the quality of our manuscript.
The comments made were very vital and we have addressed them in the manuscript and highlighted the revised areas. We have also provided a point-by-point response to all the editors and reviewers comments.

Reviewer 3 Report
Comments and Suggestions for Authors
Authors have addressed most of the points raised during the first revision, and added important missing information. Thus, the manuscript is substantially improved and after the revision of some minor points, it can be suitable for publication.
The minor points that remain unsatisfactory were highlighted in the PDF, and are presented below:
1. Authors report that the article was revised through native speakers, however, there are still typos on the text:
a. line 12: however is duplicated;
b. line 20: nine day should be replaced by nine days (plural)
c. line 24: with averages values to average values
2. Authors did not properly describe the effects of drought and their relation to osmolytes (line 92-94). In addition to the ROS-scavenging effect, they function by reducing water potential of cells, thus, maintaining water uptake even in dry soils. This information must be added, together with a reference.

Author Response

(The authors gave the same response as above.)
